# Single Nucleotide Polymorphism (SNP) Discovery and Association Study of Flowering Times, Crude Fat and Fatty Acid Composition in Rapeseed (*Brassica napus* L.) Mutant Lines Using Genotyping-by-Sequencing (GBS)

**Jaihyunk Ryu** [1], **Jae Il Lyu** [1], **Dong-Gun Kim** [1], **Kwang Min Koo** [1], **Baul Yang** [1], **Yeong Deuk Jo** [1], **Sang Hoon Kim** [1], **Soon-Jae Kwon** [1], **Bo-Keun Ha** [2], **Si-Yong Kang** [3], **Jin-Baek Kim** [1] and **Joon-Woo Ahn** [1,*]

1    Advanced Radiation Technology Institute, Korea Atomic Energy Research Institute, Jeongeup 56212, Korea; jhryu@kaeri.re.kr (J.R.); jaeil@kaeri.re.kr (J.I.L.); dgkim@kaeri.re.kr (D.-G.K.); koo@kaeri.re.kr (K.M.K.); byang@kaeri.re.kr (B.Y.); jyd@kaeri.re.kr (Y.D.J.); shkim80@kaeri.re.kr (S.H.K.); soonjaekwon@kaeri.re.kr (S.-J.K.); jbkim74@kaeri.re.kr (J.-B.K.)
2    Division of Plant Biotechnology, Chonnam National University, Gwangju 61186, Korea; bkha@chonnam.ac.kr
3    Department of Horticulture, College of Industrial Sciences, Kongju National University, Yesan 32439, Korea; sykang@kongju.ac.kr
*    Correspondence: joon@kaeri.re.kr; Tel.: +82-62-530-3314

**Abstract:** Rapeseed is the most important oil crop used in the food and biodiesel industries. In this study, based on single nucleotide polymorphism (SNP) identified from genotyping-by-sequencing (GBS), and an association study of flowering time, crude fat and fatty acid contents were investigated in 46 rapeseed mutant lines derived from gamma rays. A total of 623,026,394 clean data reads were generated with 6.6 million reads on average. A set of 37,721 filtered SNPs was used to perform gene ontology and phylogenetic analysis. Hierarchical cluster analysis of the rapeseed mutant lines gave eight groups based on flowering time and fatty acid compositions. Gene ontological analysis of the mutant lines showed that many genes displaying SNPs are involved in cellular processes, cellular anatomy, and binding. A total of 40 SNPs were significantly associated with flowering time (1 SNP), crude fat content (2 SNPs), and fatty acid content (37 SNPs). A total of 21 genes were annotated from fatty acid content SNPs; among them, nine genes were significantly enriched in reproductive processes, such as embryonic development, fruit development, and seed development. This study demonstrated that SNPs are efficient tools for mutant screening and it provides a basis that the improving the oil qualities of rapeseed.

**Keywords:** rapeseed; mutant; genotyping; genotyping-by-sequencing (GBS); association study

## 1. Introduction

Rapeseed (*Brassica napus* L.) is an oilseed plant that grows three to five feet tall and produces beautiful little yellow flowers and is widely used as food, an animal feedstock, and biodiesel. The rapeseed plant is part of the brassica family, as are cabbage, broccoli, Brussels sprouts, and mustard [1,2]. Rapeseed provides for the beautiful landscape of yellow flowers in winter or spring [2]. The plant produces pods from which seeds are harvested. Rapeseed oil is obtained from ground seeds and these seeds contain about 40% oil. Rapeseed oil is recognized as being beneficial to human health because it can lower levels of cholesterol and reduce the risk of arteriosclerosis [3,4]. Crude fat content and fatty acid composition are the characteristics of rapeseed that are important for industrial applications [3,5]. The oil qualities of rapeseed depend mainly on its fatty acid content [5,6]. Many breeders have produced new rapeseed cultivars through breeding for modified fatty acid compositions to make a specific rapeseed oil [6]. Erucic acid, which is considered unhealthy to humans, is found in rapeseed [7]. Its most serious harmful effects include

heart lesions due to insufficient oxidation [4–6]. Therefore, it was eliminated from the seed oil, which is now known as 'canola' oil. Significant increases in rapeseed oil production are due to the development of canola with very low content or traces of erucic acid [5–8]. Recently, the fatty acid content of rapeseed oil has been determined for the prospect of marketing its health benefits. Canola types of *Brassica napus* and other brassicas mainly contain palmitic (6%), stearic (2%), oleic (60%), linoleic (20%), linolenic (10%) and eicosenoic (1%) acids, which are now the dominant fatty acids in almost all rapeseed cultivars [5,8]. This fatty acid composition is considered optimal for nutritional purposes. However, a demand for rapeseed oils with other fatty acid compositions exists in the market [6–8]. High oleic acid oil (over 70%) is being popular as good source of healthy and stable vegetable oil. Furthermore, early flowering is also an important trait among rapeseed breeders to allow for expansion of its cultivation farther north, where cultivation periods are shorter [9]. The potential seed yield of rapeseed depends to a large extent on flowering time because modifications to fatty acid compositions and flowering times are major breeding goals for rapeseed [10,11].

Radiation-induced mutation is one of the most effective methods to modify plant traits, such as oil content and composition for commercialization [6,12–14]. Mutations induced by ionizing radiation range from simple base substitutions to single- and double-strand DNA breaks [12,14–16]. Changes in genome function coupled with an increase in mutation frequency are different responses of plant tissues to ionizing radiation exposure [12,15]. Gamma radiation is an ionizing radiation known to induce diverse mutational changes. Korea has limited genetic resources in its primary oil sources of rapeseed and the various methods applicable to the development of novel oil traits. Therefore, novel genetic resources are necessary to obtain various oil traits of rapeseed. Additionally, improvements in the fatty acid compositions in rapeseed that surpass natural variation have been accomplished by mutation breeding technology [12,15–19].

Next-generation sequencing (NGS) technologies have enabled efficient and economical sequencing of the plant genome and could be employed in the direct detection of single nucleotide polymorphisms (SNPs) at a genome-wide scale in plants [20,21]. GBS measures the scaled-down label of the target genome and DNA barcode adapter to arrange multiple samples simultaneously on one NGS platform using restriction enzymes [22,23]. These methods identify mutations using a condensed description of the genome, and it was developed to enable the direct study of breeding goals with high-throughput SNP genotyping analysis [23,24]. With the increasing availability of SNPs, association studies represent authoritative techniques for identifying complex quantitative traits at high resolution [25,26]. Advances in high-throughput SNP-based genotyping methods have promoted plant genomic research, facilitating faster development of markers linked to beneficial quantitative traits for use in plant breeding [23–26]. Association mapping is a procedure by which molecular markers are used for the indirect selection of useful traits in crops [27]. It has been widely applied in plant breeding for agronomical important characteristics such as yield index, flowering times, seed oil content and quality [24–27]. GBS offers a mapping tool to accelerate crop improvement [22–24]. Association mapping selection has been used to overcome some of the limitations of quantitative trait loci (QTL) mapping, especially because of the large numbers of SNP markers identified by NGS [24–26]. Association mapping is non-random association between SNP markers and agronomic or chemical characteristics in mutations in genetically diverse genotypes [21–27].

The fatty acid biosynthesis is the interaction of the biosynthetic pathway of various mechanisms in plants [28]. Acetyl-CoA is the main metabolic pathway of fatty acid chains and is involved in the synthesis of 16–18-carbon fatty acids, which are the major (up to 90%) fatty acids in rapeseed [29]. Of the different fatty acid modifications, development of high oleic acid oil is the most important one for human health [1,3,7]. Fatty acid compositions are normal quantitative traits controlled through multiple genes that regulate desaturation, and QTL for fatty acids have been mapped to rapeseed chromosomes [28–31]. Several QTL for fatty acid compositions of importance to rapeseed breeding programs have been

successfully used in association mapping selection [2,9–11]. Additionally, many QTL for flowering time have been reported in rapeseed that co-localize with yield and major QTL that explain the variation accounting flowering time [9,25,32].

We have developed mutant rapeseed lines derived from gamma-radiation mutation, and these lines have a variety of flowering times and increased crude fat and fatty acid contents. The objectives of this study were the identification of SNP in gamma-irradiated rapeseed mutant lines. Meanwhile, new SNP locations for flowering time, crude fat content, and fatty acid composition in rapeseed mutant lines were found by an association study using GBS analysis.

## 2. Materials and Methods

### 2.1. Plant Materials

The mutant lines of rapeseed were generated by the treatment of seeds of the commercial cultivar 'Tammi' with 500 Gy of gamma ($^{60}$Co) radiation at 2009. The development of the mutant lines of rapeseed is shown in Figure 1. The selfing procedure was continued until the $M_9$ generations. Mutants that varied in flowering time, crude fat content, and fatty acid content that exhibited stable inheritance of the mutated characteristics from $M_5$ to $M_7$ generations were selected. The radiation-generated mutant genotypes were grown by the Radiation Breeding Research Team at the Advanced Radiation Technology Institute of the Korea Atomic Energy Research Institute. Young leaves were sampled from the original cultivar 'Tammi' and the 46 rapeseed mutants. Genomic DNA was isolated using a DNeasy Plant Mini Kit (Qiagen, Hilden, Germany).

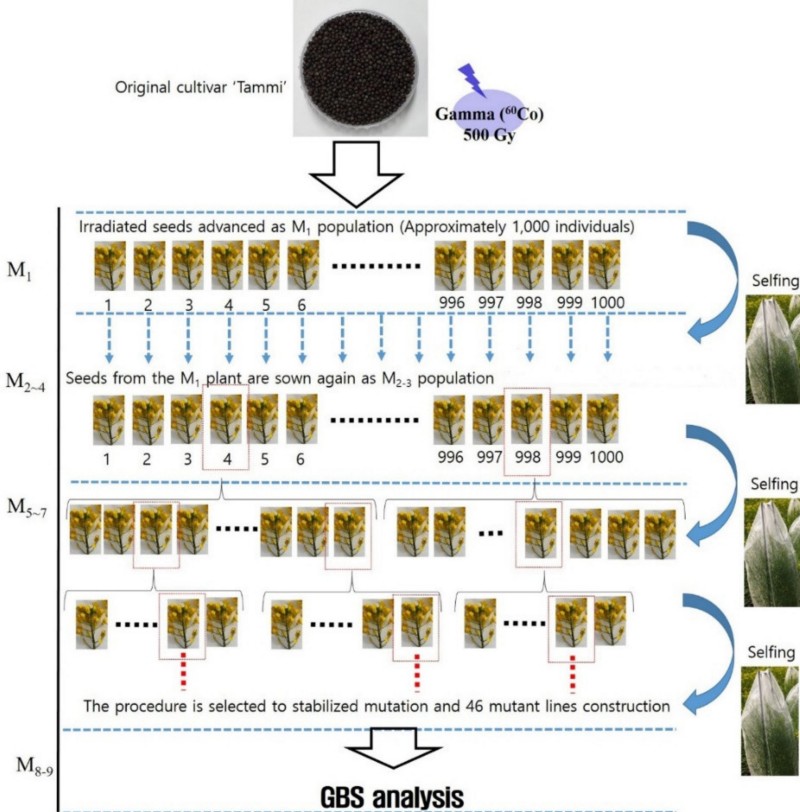

**Figure 1.** The development of mutant lines. The 46 mutant lines were identified from the $M_2$ to the $M_4$ population that flowering times, crude fat and fatty acid composition and was derived from the Korean rapeseed cultivar 'Tammi' with 500 Gy gamma ray irradiation. This mutant line was homozygous from the $M_5$ to the $M_{8-9}$ generations. At each generation, phenotypes, crude fat, and fatty acid compositions were screened.

## 2.2. Library Construction and Genotypin-by-Sequencing

GBS libraries were constructed using the restriction enzyme *Ape*KI (5′-GCWGC-3′) following a protocol modified from that of previous study [23]. Oligonucleotides constituting the top and bottom strands of each barcode adapter and a common adapter were diluted (separately) with TE (50 μM each), and annealed using a thermocycler. DNA samples (100 ng/μL) were added to individual adapter-containing wells. Samples (DNA with adapters) were digested with *Ape*KI (New England Biolabs, Ipswich, MA, USA) overnight at 75 °C. Sets of digested DNA samples, each with a different barcode adapter, were combined (5 μL each), and purified using a commercial kit (QIAquick PCR Purification Kit; Qiagen, Valencia, CA, USA), according to the manufacturer's instructions. Restriction fragments from each library were then amplified in 50 μL volumes containing 2 μL of pooled DNA fragments, HerculaseII Fusion DNA Polymerase (Agilent, CA, USA), and 25 pmol each of the following primers: (A) 5′-AAT GAT ACG GCG ACC ACC GAG ATC TAC ACT CTT TCC CTA CAC GAC GCT CTT CCG ATC T-3′ and (B) 5′-CAA GCA GAA GAC GGC ATA CGA GAT CGG TCT CGG CAT TCC TGC TGA ACC GCT CTT CCG ATC T-3′. These amplified sample pools constituted a sequencing "library". The library was sequenced on the Illumina Hiseq 2000 platform by SEEDERS Co. (Daejeon, Korea). GBS raw reads data are available in NCBI Sequence Read Archive (SRA) under the accession SRR13575739.

## 2.3. Sequence Preprocessing and Alignment to Reference Genome Sequence

Demultiplexing was performed using the barcode sequence, and adapter sequence removal and sequence quality trimming were performed. Adapter trimming was performed using cutadapt (version 1.8.3) [33], and sequence quality trimming was performed using DynamicTrim and LengthSort of the SolexaQA program (v.1.13) [20]. DynamicTrim cuts low-quality bases at both ends of short reads according to the Phred score, and refines it with high-quality cleaned reads. LengthSort removes excess base cuts made in DynamicTrim; Phred score of Dynamic-Trim $\geq$ 20, and LengthSort using short read lengths $\geq$ 25 bp. BWA (0.6.1-r104) [34] generated cleaned reads passing the preprocessing process, and performed mapping to the reference genome sequence (*Brassica napus* V5.1; (http://www.genoscope.cns.fr/brassicanapus accessed on 5 February 2021). Mapping was a preliminary step to detect raw SNPs (In/Del) between *B. napus* V5.1 and sequenced samples. A SAM file was created, and default values were used, except for the following options: a seed length ($-l$) of 30, maximum differences in the seed ($-k$) of 1, number of threads ($-t$) of 16, mismatch penalty ($-M$) of 6, gap opening penalty ($-O$) of 15, and gap extension penalty (-$E$) of 8. The experiment was conducted in repetition.

## 2.4. Raw SNP Detection and Consensus Sequence Extraction

Clean reads were mapped to the standard genome sequence, and the generated SAM files were used to detect raw SNPs using SAMtools (0.1.16) [35], and extract consensus sequences. SNP validation was performed using SEEDERS in-house script [36] before SNP detection; raw SNP detection was performed, and default values were used, except for the following options: a minimum mapping quality for SNPs ($-Q$) of 30, minimum mapping quality for gaps ($-q$) of 15, minimum read depth ($-d$) of 3, minimum indel score for nearby SNP filtering ($-G$) of 30, SNPs within INT bp around a gap to be filtered ($-w$) of 15, window size for filtering dense SNPs ($-W$) of 30, and maximum read depth ($-D$) of 223.

## 2.5. Generate SNP Matrix

To conduct the analysis of SNPs between the analyzed objects, an integrated SNP matrix was generated between samples. A list of unions was created using the raw SNP positions obtained by comparing each sample with a standard dielectric, and a non-SNP locus was filled in from the consensus sequence of the sample. Then, the final SNP matrix was generated by filtering the miscalled SNP positions through SNP comparison among samples. SNPs were divided into homozygous (SNP read depth $\geq$ 90%), heterozygous (40% $\leq$ SNP read depth $\leq$ 60%), etc., (homozygous/heterozygous; could not be distinguished

by type) groups based on their position. The named SNP positions were classified into "intergenic or genic regions" based on the position information of the standard genome sequence (*Brassica napus* V. 5.1), and the genic region was further classified into "CDS or intron regions." The common SNP in original cultivar 'Tammi' was selected first in the integrated SNP matrix position between mutants, and the polymorphic SNP was selected by comparing the common SNP of the original cultivar with the base sequence of each mutant. In order to perform GO analysis and flexible relationship analysis, SNP loci of each mutant line are integrated to secure the SNP locus of the union.

### 2.6. Gene Ontology Analysis of Genes with Polymorphic SNPs

Gene ontology alignment was performed using candidate sequences containing polymorphic SNPs and sequences provided by GO database in house scrips [37]. Thresholds were classified into three functional categories: BP (Biological Process), CC (Cellular Component), and MF (Molecular Function), with the significance level for 0.01 (e-value $\leq 1.0 \times 10^{-10}$, best hits).

### 2.7. Construction of Phylogenetic Tree and Heatmap

The evolutionary history of the SNP was inferred using the neighbor-joining method. The optimal tree with the sum of branch lengths equal to 2.32462671 is shown. The percentage of replicate trees in which the same clusters were formed, determined by bootstrapping analysis (1000 replicates), are shown next to the branches. The tree is drawn to scale, with branch lengths in the same units as those of the evolutionary distances used to infer the phylogenetic tree. The evolutionary distances were computed using the maximum composite likelihood method and are in units of number of base substitutions per site. The analysis involved 47 nucleotide sequences. All ambiguous positions were removed for each sequence pair. A total of 35,397 positions were included in the final dataset. Evolutionary analyses were conducted using MEGA6 [38]. The clustering analysis of samples from the 47 rapeseed lines was performed using the complete linkage method in the SPSS (IBM, Armonk, NY, USA). The flowering times, crude fat contents, and fatty acid compositions were visualized as z-values on the heatmap. The hierarchical clustering analysis of the 35,397 union SNPs was performed using the neighbor-joining method in TASSEL v5.0.

### 2.8. Association Analysis

Association analysis was performed using 35,397 union SNP dataset and FarmCPU model in genomic association and prediction integrated tool (GAPIT) R package [39]. The retained PC obtained in association analysis was used as a covariate in the FarmCPU procedure, as well as also kinship matrix. Significant SNPs were defined based on the threshold of $p < 0.0001$. The annotated genes as selected significant SNPs were evaluated enrichment analysis of GOBP category for defined gene functions by AgriGO database ( http://bioinfo.cau.edu.cn/agriGO/ accessed on 5 February 2021) based on the threshold of Hochberg multitest adjustment method (false discovery rate (FDR) < 0.05).

### 2.9. Determination of Crude Fat and Fatty Acid Compositions

The seed oil content was analyzed using the AOAC method as previously described [40]. Using the Soxhlet extraction procedure, 5 g crushed seeds (80 mashed) were packed into a thimble and the oils were extracted with diethyl ether for 6 h. Fatty acid composition was measured previously described method [17]. The rapeseed oil was extracted by rapeseed powder in chloroform: hexane: methanol (8:5:2, *v/v/v*) 1 mL for 12 h. From this, 200 μL of extracted oil was added to 75 μL of methylation reagent (0.25 M methanolic sodium methoxide: petroleum ether: ethyl ether = 1:5:2, *v/v/v*) for derivatization. Hexane was added to bring the total volume up to 1 mL. The fatty acid composition of the rapeseed seed oil was analyzed using a GC-MS (Plus-2010, Shimadzu, Japan) instrument equipped with an HP-88 capillary column (J&W Scientific, 60 m $\times$ 0.25 mm $\times$ 0.25 m) under the following conditions: ionization voltage, 70 eV; mass scan range, 50–450 mass units; injector

temperature, 230 °C; detector temperature, 230 °C; injection volume, 1 µL; split ratio, 1:30; carrier gas, helium; and flow rate, 1.7 mL/min. The column temperature program specified an isothermal temperature of 40 °C for 5 min increasing to 180 °C at a rate of 5 °C/min then a subsequent increase to 230 °C at a rate of 1 °C/min. We identified the substances present in the extracts according to their retention time (RT) and using the mass spectra database (NIST 62 Library).

## 3. Results

*3.1. Flowering Time, Crude Fat Content, and Fatty Acid Compositions between the Original Cultivar and Mutant Lines*

The flowering times, crude fat contents, and fatty acid compositions of the 46 mutant lines and the original cultivar ('Tammi') are shown in Table 1. The 47 rapeseed lines were divided into three categories: early flowering (15 mutant lines), intermediate flowering time (original cultivar and 26 mutant lines), and late flowering (6 mutant lines). The crude fat content of the original cultivar was 33.9 mg/100 g. The crude fat content for all mutant lines ranged from 26.9 to 42.3 mg/100 g with an average of 34.4 mg/100 g. The Tm10-1, Tm8-2, and Tm8-16 lines contained higher levels of crude fat than other rapeseed lines.

**Table 1.** Flowering time, crude fat content, and fatty acid compositions of rapeseed mutant used in this study.

| Lines | Flowering Types | Crude Fat Content (mg/100 g) | Fatty Acid Composition (%) | | | | | |
|---|---|---|---|---|---|---|---|---|
| | | | C16:0 | C18:0 | C18:1 | C18:2 | C18:3 | C22:1 |
| Tammi | Middle * | 33.92 | 4.8gh ** | 1.9c | 64.9d | 15.7e | 4.4e | 0.0f |
| Tm2M-1 | Late | 37.29 | 3.6k | 0.4f | 70.7bc | 17.1e | 7.2d | 0.0f |
| Tm3M-1 | Late | 37.11 | 4.9g | 0.9e | 68.7c | 16.7e | 7.7cd | 0.0f |
| Tm3M-2 | Late | 38.11 | 2.9m | 0.2g | 70.1bc | 18.8d | 7.1de | 0.0f |
| Tm4M-1 | Late | 37.73 | 2.3n | 0.6f | 69.8bc | 20.3cd | 6.4de | 0.0f |
| Tm4M-2 | Early | 36.47 | 4.4i | 1.7d | 69.7c | 14.5f | 5.8d | 0.0f |
| Tm7M-1 | Late | 37.95 | 4.8gh | 1.6d | 64.3d | 16.8e | 5.7d | 0.0f |
| Tm7M-2 | Late | 36.10 | 4.0jk | 0.0h | 71.6b | 13.5e | 4.5e | 0.0f |
| Tm6-1 | Early | 36.23 | 4.9g | 0.6f | 67.9cd | 18.6de | 7.2d | 0.0f |
| Tm6-2 | Middle | 38.50 | 5.6f | 0.0h | 67.8cd | 17.8e | 5.4de | 0.0f |
| Tm6-3 | Early | 34.96 | 4.5hi | 2.3b | 67.4cd | 16.7e | 5.9d | 0.0f |
| Tm6-4 | Middle | 35.95 | 4.0jk | 0.0h | 72.1b | 15.5e | 4.7e | 0.1e |
| Tm6-6 | Middle | 33.42 | 5.5f | 1.9c | 70.3bc | 17.0e | 4.7e | 0.0f |
| Tm6-7 | Middle | 32.50 | 3.5kl | 0.0h | 71.3bc | 15.0ef | 6.1d | 0.0f |
| Tm6-8 | Middle | 27.13 | 3.3l | 0.5f | 76.7a | 14.6f | 4.2e | 0.0f |
| Tm6-10 | Middle | 32.69 | 8.2c | 1.8cd | 49.8h | 27.0a | 10.9b | 0.0f |
| Tm6-12 | Middle | 35.21 | 5.3fg | 1.1e | 62.0e | 22.5c | 8.0c | 0.0f |
| Tm6-13 | Middle | 29.72 | 2.5 | 0.8ef | 76.1a | 14.9ef | 5.2de | 0.0f |
| Tm8-2 | Middle | 40.46 | 3.4l | 0.0e | 72.0b | 13.5f | 5.7d | 0.0f |
| Tm8-3 | Middle | 32.65 | 4.4i | 1.0e | 68.4c | 17.0e | 8.0cd | 0.0f |
| Tm8-4 | Middle | 38.09 | 6.8e | 1.8cd | 52.0g | 28.3a | 10.0b | 0.0f |
| Tm8-5 | Early | 37.04 | 4.8gh | 0.3g | 59.1ef | 26.6ab | 8.3cd | 0.0f |
| Tm8-6 | Middle | 33.69 | 7.4de | 0.4f | 51.7g | 28.4a | 11.3ab | 0.0f |
| Tm8-7 | Middle | 37.62 | 9.2b | 0.8ef | 48.6h | 28.8a | 11.6ab | 0.0f |
| Tm8-8 | Middle | 38.29 | 2.5n | 0.3g | 65.4d | 26.6ab | 4.9e | 0.0f |
| Tm8-10 | Middle | 33.25 | 4.4i | 2.4ab | 57.6f | 21.7cd | 6.0d | 0.0f |
| Tm8-11 | Middle | 34.19 | 5.1g | 2.7a | 56.7f | 21.0cd | 4.6e | 0.0f |
| Tm8-12 | Middle | 36.54 | 4.7h | 2.1bc | 56.3f | 20.7cd | 5.8d | 0.0f |
| Tm8-13 | Middle | 30.15 | 4.6hi | 2.2b | 59.8e | 20.7cd | 5.1de | 0.0f |
| Tm8-14 | Middle | 32.45 | 5.4f | 2.4ab | 51.6gh | 14.8f | 1.4f | 2.0d |
| Tm8-15 | Middle | 33.18 | 3.5kl | 1.0e | 37.3i | 21.3cd | 6.3de | 15.8b |
| Tm8-16 | Middle | 41.32 | 3.7k | 2.2b | 62.7de | 18.8d | 5.0e | 0.0f |
| Tm8-17 | Middle | 38.29 | 4.1jk | 2.3b | 59.5e | 20.5cd | 5.8d | 0.0f |
| Tm10-1 | Early | 42.32 | 1.3p | 0.2g | 71.5b | 21.9cd | 4.9e | 0.0f |
| Tm10-1St | Early | 28.15 | 2.5n | 0.2g | 35.3i | 18.7de | 8.8c | 20.1a |

**Table 1.** *Cont.*

| Lines | Flowering Types | Crude Fat Content (mg/100 g) | Fatty Acid Composition (%) | | | | | |
|---|---|---|---|---|---|---|---|---|
| | | | C16:0 | C18:0 | C18:1 | C18:2 | C18:3 | C22:1 |
| Tm10-1Lin | Middle | 38.16 | 9.8a | 1.8cd | 48.8h | 24.9bc | 12.8a | 0.0f |
| Tm10-2 | Middle | 38.10 | 9.1b | 0.4f | 53.1g | 25.4b | 10.6b | 0.0f |
| Tm10Oel | Early | 31.85 | 3.3l | 0.4f | 70.5bc | 19.1d | 6.0d | 0.0f |
| Tm10-3 | Middle | 33.74 | 6.9e | 1.7d | 57.2f | 24.2bc | 8.4c | 0.0f |
| Tm10-4EF | Early | 36.49 | 5.4f | 1.6d | 60.8e | 23.7c | 7.8cd | 0.0f |
| Tm10-5EF | Early | 26.85 | 1.7o | 0.2g | 71.4b | 22.3c | 4.0e | 0.0f |
| Tm10-6EF | Early | 28.44 | 3.7k | 0.0h | 56.6f | 17.1e | 5.9d | 5.9c |
| Tm10-7EF | Early | 26.92 | 3.5kl | 1.9c | 63.5d | 19.2d | 4.8e | 0.0f |
| Tm10-8EF | Middle | 29.46 | 3.5kl | 0.0h | 64.2d | 19.1d | 8.6c | 0.1e |
| Tm10-9EF | Early | 28.55 | 3.4l | 0.0h | 59.6e | 19.2d | 6.7de | 4.8c |
| Tm10-10EF | Early | 29.98 | 3.8k | 0.0h | 69.4bc | 17.7e | 5.0e | 0.1e |
| Tm10-11EF | Early | 27.99 | 3.6k | 0.0h | 68.2c | 18.9d | 5.1de | 0.2e |

* Early: The day to flowering were ranged from 160 to 169 days after sowing; Middle: The day to flowering were ranged from 170 to 179 days after sowing; Late: The day to flowering were more than 180 days after sowing. C16:0-palmitic acid, C18:0-stearic acid, C18:1-oleic acid, C18:2-linoleic acid, C18:3-linolenic acid, C22:1-erucic acid. ** The letters above each point indicate a significant difference at the 0.05 probability level (Duncan's multiple range tests, n = 3).

The fatty acid composition of the original cultivar was 4.8%, 1.9%, 64.9%, 15.7%, and 4.4% for palmitic, stearic, oleic, linoleic, and linolenic acids, respectively. Oleic, linoleic, and linolenic acids were the principal fatty acids represented in the rapeseed lines except for three mutant lines. Oleic, linoleic, and erucic acids were the major fatty acids in the Tm4M-2, Tm8-15, and Tm10-1St mutant lines. The palmitic acid content of the mutant lines ranged from 1.3% to 9.8%, with an average of 4.6%. Stearic acid was detected in 36 mutant lines, with a composition ranging from 0.2% to 2.7%. Significant differences in oleic acid composition were observed between all the mutant lines. The oleic acid contents of the mutant lines ranged from 35.3% to 76.7%. The Tm6-8 mutant line had the highest oleic acid content, whereas the Tm10-1St mutant line had the lowest. The highest linoleic acid content was recorded at 28.8% in the Tm8-7 mutant line, and the lowest value of 13.5% was found in the Tm8-2 and Tm7M-2 lines. The highest content of linolenic acid was found in the Tm10-1Lin line (12.8%), while the lowest content was detected in the TM8-14 line (1.4%). Erucic acid was determined in nine mutant lines (Tm8-14, Tm8-15, Tm10-1St, Tm10-6EF, Tm10-8EF, Tm10-9EF, Tm10-10EF, and Tm10-11EF) with a composition range of 0.1% to 21.6%. The Tm10-1St mutant line had the highest erucic acid content.

### 3.2. Genotyping-by-Sequencing of Rapeseed Mutant Lines

The GBS library was constructed from 46 rapeseed mutant lines, and the original cultivar was sequenced using the Illumina Hiseq 2000 platform (Illumina, Madison, WI, USA). A summary of these sequencing results is presented in Table 2 and Table S1. GBS was carried out a second time (two biological replicates were used for GBS analysis), and a total of 710 million reads comprising 107,525,459,536 nucleotides (107.5 Gb) were generated, with an average of 7.5 million reads per genotype. After removing low quality sequences, 623,026,394 clean reads remained, with 6.6 million reads per genotype on average. The total length range of clean reads was between 70,671,425 bp and 1,768,913,538 bp, with an average read length of 646,640,540 bp. The total number of mapped reads was 396,325,056 in all lines, with an average of 4,216,224 reads per sample. The mapped read rates (%) ranged from 29.42% to 85.14%. On average, 54.6% of the filtered reads were mapped to the reference genome sequence. The total length of the mapped region was 1,537,448,005 bp, with an average of 16,355,830 bp per sample, which covered approximately 1.92% of the reference genome sequence. Among the 47 lines, the average depth of the mapped region ranged from 7.83 to 51.44 (Table S1).

**Table 2.** Summary of GBS sequence data and alignment to the reference genome sequence.

|  | Total | Average/Plant |
|---|---|---|
| Raw data | | |
| Reads | 712,089,136 | 7,575,416 |
| Bases (bp) | 107,525,459,536 | 1,143,887,867 |
| After trimming | | |
| Reads | 623,026,394 | 6,627,540 |
| Bases (bp) | 60,784,210,724 | 646,640,540 |
| Mapped reads on reference genome * | | |
| Reads | 396,325,056 | 4,216,224 |
| Bases (bp) | 1,537,448,005 | 16,355,830 |
| Reference genome coverage (%) | | 1.924% |

* Reference genome sequence: *Brassica napus* V5.1 (http://www.genoscope.cns.fr/brassicanapus accessed on 5 February 2021).

### 3.3. Identification of SNPs

The SNPs for each line were first selected from the union of SNPs in the matrix position between samples and the reference genome sequence (Table S2). A total of 1,528,875 SNPs were identified for all lines, of which the largest number of SNPs was recorded for the Tm8-10 mutant line and the lowest number was observed in the Tm7-1EF mutant line. To identify gamma-irradiated mutant lines, the polymorphic SNPs collected by comparing the common SNPs (1st selected 1,528,875 SNPs) in the original cultivar with the base sequences of the mutant lines were considered mutational changes (Table S3 and Supplementary Materials File S1). A total of 277,036 SNPs were observed in 46 mutants. Most of the SNPs (203,046) were homozygous, and there were 73,990 heterozygous variants. A union of 35,397 SNPs without overlap were constructed by combining SNPs for each mutant line. The SNP distribution in the rapeseed chromosomes ranged from 229 to 2765, with a mean of 1605 per chromosome (Figure 2). Specifically, chromosomes C01, C03, A09, and A03 had the highest numbers of SNPs.

Functional annotation of the SNPs in gamma-irradiated mutant lines was performed using the reference genome sequence (Table 3). Of the 35,397 union SNPs, 21,697 SNPs (61.3%) were located in the genic region, and 13,700 (38.7%) were detected in the intergenic region, encompassing 10,834 genes. Within the genic region, most of the SNPs were in untranslated regions (UTRs, 15,651 SNPs), while the remaining SNPs were located in coding sequences (CDS, 4657 SNPs), promoters (2499 SNPs), and introns (460 SNPs).

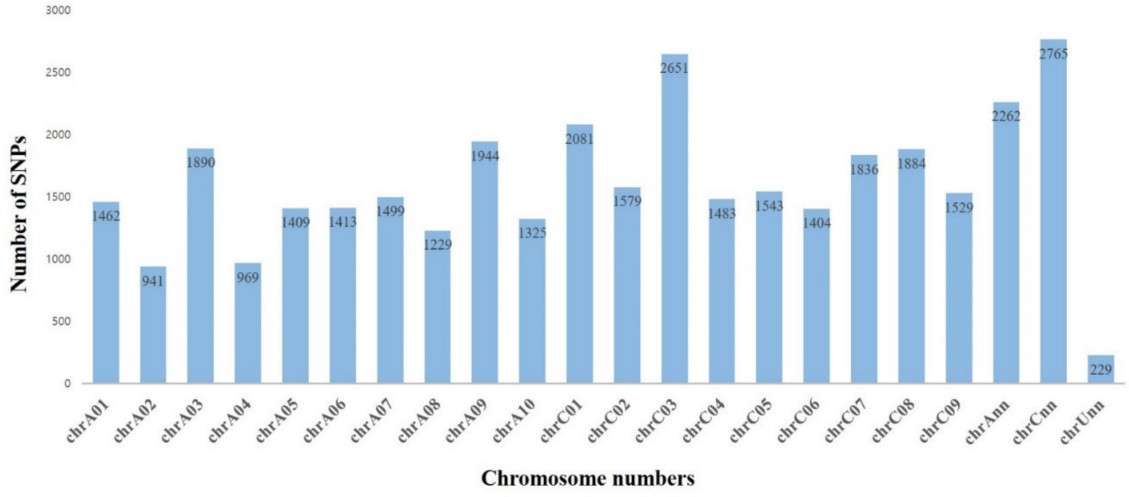

**Figure 2.** Chromosomal distribution of the 35,397 union SNPs from GBS in gamma-irradiated rapeseed mutant lines. The union SNPs were combined from the 277,036 mutational changed SNPs. Reference genome sequence: *Brassica napus* V5.1 (http://www.genoscope.cns.fr/brassicanapus).

**Table 3.** Statistics of polymorphic single nucleotide polymorphisms (SNPs) annotation by genomic locations.

| No. | Line Names | Polymorphic SNPs | Polymorphic SNPs | | Genic Region | | | | Genes Number |
|---|---|---|---|---|---|---|---|---|---|
| | | | Intergenic Region | Genic Region | Promoter 1kb | UTR | CDS | Intron | |
| 1 | Tm2M-1 | 3335 | 1341 | 1994 | 220 | 1458 | 394 | 53 | 1244 |
| 2 | Tm3M-1 | 2766 | 1074 | 1692 | 198 | 1251 | 331 | 36 | 1040 |
| 3 | Tm3M-2 | 5402 | 2022 | 3380 | 381 | 2467 | 670 | 98 | 1895 |
| 4 | Tm4M-1 | 4338 | 1661 | 2677 | 291 | 1964 | 538 | 67 | 1575 |
| 5 | Tm4M-2 | 6276 | 2233 | 4043 | 436 | 2961 | 829 | 92 | 2345 |
| 6 | Tm7M-1 | 2376 | 539 | 1837 | 185 | 1361 | 375 | 41 | 1118 |
| 7 | Tm7M-2 | 5630 | 2133 | 3497 | 392 | 2537 | 703 | 98 | 1940 |
| 8 | Tm6-1 | 6164 | 1602 | 4562 | 465 | 3352 | 987 | 90 | 2461 |
| 9 | Tm6-2 | 3990 | 1258 | 2732 | 288 | 2036 | 555 | 58 | 1648 |
| 10 | Tm6-3 | 5983 | 1987 | 3996 | 430 | 2895 | 866 | 83 | 2173 |
| 11 | Tm6-4 | 5016 | 1395 | 3621 | 364 | 2705 | 732 | 66 | 2061 |
| 12 | Tm6-6 | 7342 | 2882 | 4460 | 483 | 3208 | 947 | 117 | 2467 |
| 13 | Tm6-7 | 5176 | 1498 | 3678 | 376 | 2734 | 757 | 73 | 2030 |
| 14 | Tm6-8 | 6265 | 1764 | 4501 | 494 | 3334 | 946 | 84 | 2470 |
| 15 | Tm6-10 | 6342 | 2293 | 4049 | 452 | 2923 | 839 | 106 | 2327 |
| 16 | Tm6-12 | 5459 | 1601 | 3858 | 456 | 2820 | 786 | 99 | 2121 |
| 17 | Tm6-13 | 7045 | 2499 | 4546 | 460 | 3300 | 983 | 106 | 2673 |
| 18 | Tm8-2 | 8628 | 2671 | 5957 | 657 | 4292 | 1289 | 143 | 3364 |
| 19 | Tm8-3 | 6100 | 2254 | 3846 | 459 | 2818 | 781 | 80 | 2144 |
| 20 | Tm8-4 | 8732 | 2714 | 6018 | 647 | 4382 | 1295 | 123 | 3222 |
| 21 | Tm8-5 | 5614 | 1872 | 3742 | 400 | 2762 | 782 | 73 | 2063 |
| 22 | Tm8-6 | 6372 | 1675 | 4697 | 496 | 3463 | 983 | 90 | 2640 |
| 23 | Tm8-7 | 6183 | 1800 | 4383 | 474 | 3242 | 926 | 84 | 2389 |
| 24 | Tm8-8 | 8315 | 2805 | 5510 | 610 | 4012 | 1178 | 118 | 3002 |
| 25 | Tm8-10 | 7811 | 2361 | 5450 | 643 | 3891 | 1222 | 129 | 3045 |
| 26 | Tm8-11 | 9049 | 2491 | 6558 | 713 | 4761 | 1425 | 143 | 3504 |
| 27 | Tm8-12 | 8707 | 2810 | 5897 | 642 | 4332 | 1257 | 112 | 3169 |
| 28 | Tm8-13 | 9446 | 2723 | 6723 | 720 | 4937 | 1425 | 140 | 3566 |
| 29 | Tm8-14 | 9024 | 2888 | 6136 | 684 | 4471 | 1317 | 128 | 3328 |
| 30 | Tm8-15 | 8491 | 2548 | 5943 | 662 | 4341 | 1265 | 139 | 3245 |
| 31 | Tm8-16 | 5418 | 1583 | 3835 | 406 | 2821 | 811 | 71 | 2273 |
| 32 | Tm8-17 | 8707 | 2885 | 5822 | 667 | 4208 | 1258 | 132 | 3214 |
| 33 | Tm10-1 | 5717 | 1885 | 3832 | 414 | 2777 | 834 | 75 | 2207 |
| 34 | Tm10-1St | 6593 | 1936 | 4657 | 509 | 3388 | 1016 | 88 | 2690 |
| 35 | Tm10-1Lin | 4887 | 1403 | 3484 | 368 | 2592 | 719 | 66 | 2013 |
| 36 | Tm10-2 | 3754 | 907 | 2847 | 318 | 2123 | 566 | 53 | 1692 |
| 37 | Tm10Oel | 5243 | 1670 | 3573 | 385 | 2643 | 706 | 85 | 2098 |
| 38 | Tm10-3 | 2891 | 852 | 2039 | 213 | 1527 | 421 | 30 | 1248 |
| 39 | Tm10-4EF | 5269 | 1725 | 3544 | 320 | 2612 | 753 | 70 | 2228 |
| 40 | Tm10-5EF | 5684 | 1644 | 4040 | 429 | 3002 | 840 | 71 | 2250 |
| 41 | Tm10-6EF | 4827 | 1480 | 3347 | 352 | 2459 | 698 | 73 | 1954 |
| 42 | Tm10-7EF | 4856 | 1620 | 3236 | 360 | 2402 | 643 | 67 | 1888 |
| 43 | Tm10-8EF | 5094 | 1751 | 3343 | 366 | 2497 | 648 | 68 | 1943 |
| 44 | Tm10-9EF | 7490 | 2399 | 5091 | 585 | 3708 | 1071 | 110 | 2966 |
| 45 | Tm10-10EF | 5538 | 1509 | 4029 | 420 | 2984 | 852 | 83 | 2238 |
| 46 | Tm10-11EF | 3691 | 1034 | 2657 | 271 | 1927 | 583 | 63 | 1576 |
| | Union SNP | 35,397 | 13,700 | 21,697 | 2499 | 15,651 | 4657 | 460 | 10,834 |

### 3.4. GO Analysis of Genes with Polymorphic SNPs

For the functional classification of the genes mutated by gamma rays in the mutant lines, gene ontology (GO) enrichment analysis was conducted with the 10,834 genes (Table S4) carrying polymorphic SNPs ($p < 0.01$). The genes were classified into three main categories: biological process (BP), cellular component (CC), and molecular function (MF) genes (Figure 3). Genes containing BP SNPs included those under cellular processes (6431 genes), primary metabolic processes (5159 genes), nitrogen compound metabolic processes (4472 genes), and macromolecule metabolic processes (4285 genes). CC SNPs included intracellular entities (8046 genes), organelle entities (7152 genes), and intracellular organelles (7130 genes). MF included nucleotide binding (1564 genes), nucleoside phosphate binding (1564), anion binding (1429 genes), carbohydrate derivative binding (1217 genes), and purine nucleotide binding (1214 genes).

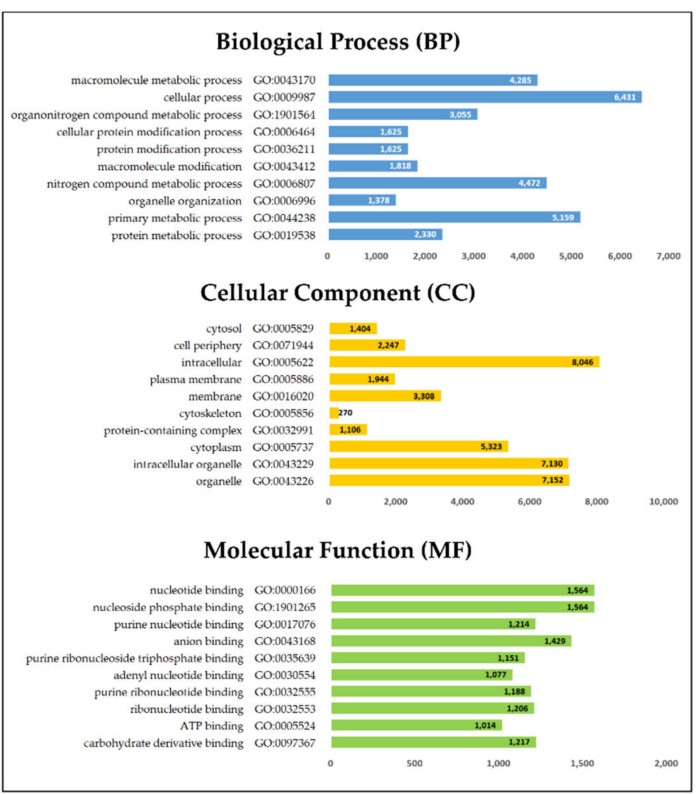

**Figure 3.** Histogram of GO terms of union SNPs in rapeseed mutant lines.

### 3.5. Phylogenetic and Hierarchical Cluster Analysis

A phylogenetic analysis was carried out between the original cultivar and the 46 mutant lines based on the maximum composite likelihood method, and a dendrogram was generated with 35,397 union SNPs using the neighbor-joining method (Figure 4). The cluster analysis suggested that the mutant genotypes could be divided into eight related groups and two independent groups: Group I contained nine mutants (Tm10Oel, Tm2M-1, Tm3M-1, Tm3M-2, Tm4M-1, Tm4M-2, Tm6-6, Tm7M-1, and Tm7m-2), and Group II included two early flowering mutants, Tm10-7 and Tm10-8EF. Group III contained the original cultivar and the Tm6-12 line. Group IV consisted of the Tm8-3 and Tm6-3 mutant lines. Group V contained four mutant lines (Tm6-13, Tm10-1st, Tm10-4, and Tm10-9). Group VI contained five mutant lines (Tm6-2, Tm6-4, Tm10-2, Tm10-3, and Tm10-11EF). Eight mutant lines (Tm6-1, Tm6-7, Tm6-8, Tm8-7, Tm10-1Lin, Tm10-5EF, Tm10-6EF, and Tm10-10EF) belonged to Group VII. Group VIII contained 12 mutant lines (Tm8-10, Tm8-11, Tm8-12, Tm8-13, Tm8-14, Tm8-15, Tm8-16, Tm8-17, Tm8-2, Tm8-4, Tm8-6, and Tm8-8). Three mutant lines (Tm10-1, Tm6-10, and Tm8-5) did not belong to any groups.

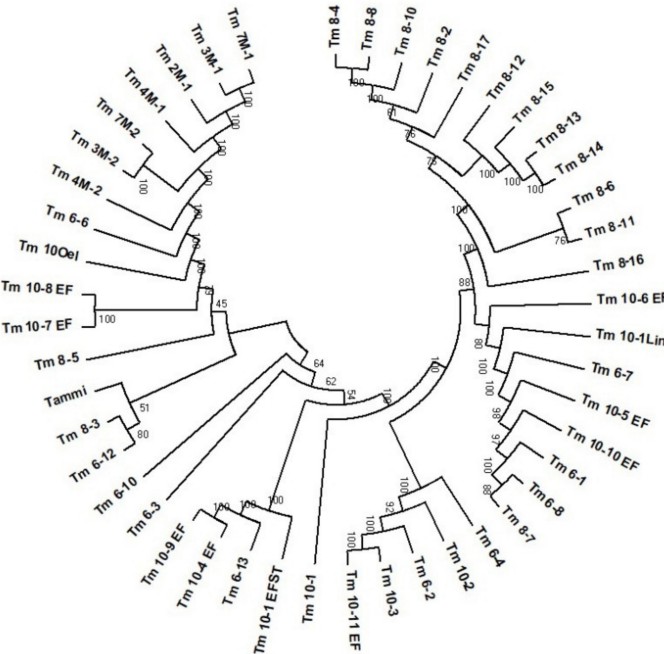

**Figure 4.** Neighbor-joining dendrograms based on pairwise distance matrix representing the grouping of the 46 rapeseed mutant lines and original cultivar obtained from 35,397 union SNPs from GBS.

The results of the flowering times, crude fat contents, and fatty acid compositions hierarchical cluster analysis are presented in Figure 5. The original cultivar and the 46 mutant lines were clustered into 10 groups and one independent group. Group I contained four late, three intermediate, and one early flowering time lines with oleic acid contents ranging from 68.4% to 70.7%. Group II consisted of the original cultivar and four mutant lines with oleic acid contents ranging from 64.9% to 67.4%. Group III included three mutants with high oleic and erucic acid contents. Group IV contained two mutant lines with high crude fat and linoleic acid contents. Group V had five mutant lines with low crude fat and high oleic acid contents. Group VI possessed six mutant lines with low oleic acid and high palmitic, linoleic acid, and linolenic acid contents. Group VII consisted of two mutant lines with low oleic acid and high crude fat contents. Seven mutant lines showing high linoleic acid and intermediate crude fat contents belonged to Group VIII. Group IX contained five mutant lines with low crude fat and intermediate oleic and linoleic acid contents. Two mutant lines with low crude fat and high erucic acid contents belonged to Group X. The Tm8-14 line did not belong to any groups. Hierarchical cluster analysis divided flowering time, crude fat content, and six fatty acid compounds into two clusters and two independent groups. Cluster I contained three fatty acid compounds: palmitic, linoleic, and linolenic acid. Cluster II contained stearic acid, crude fat content, and flowering time. Oleic and erucic acid contents formed independent groups.

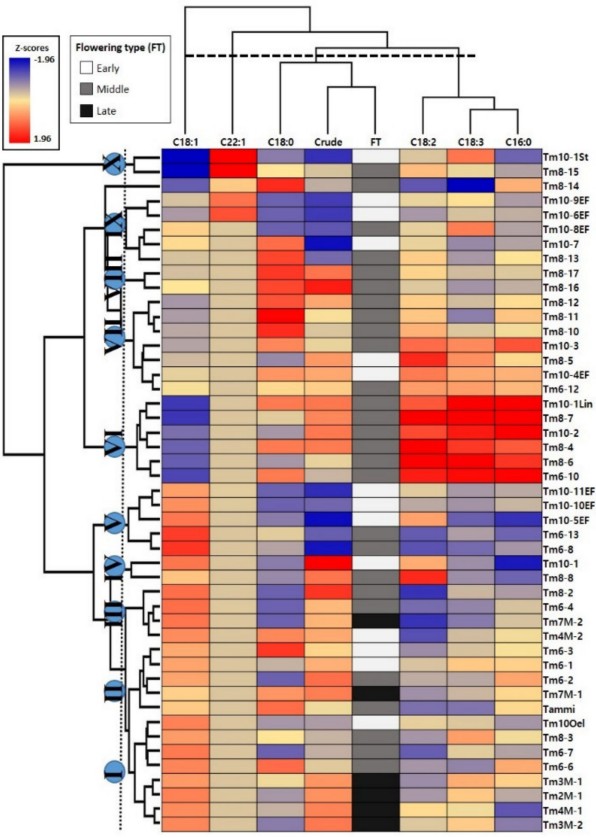

**Figure 5.** Hierarchical cluster analysis of the 47 rapeseed lines according to their flowering time, crude fat, and fatty acid content.

### 3.6. Association Analysis via GBS

All SNPs with significantly associated with flowering time, crude fat content, and fatty acid composition are described in Table 4. Among all 35,397 union SNP dataset associations identified, 40 SNPs were significantly associated with fatty acid composition, flowering time, and crude fat content by applying the fixed and random model circulating probability unification (FarmCPU). Of the 40 selected union SNPs, 28 SNPs were located in genic regions, and 12 SNPs were detected in intergenic regions. A total of 37 SNP loci were significantly associated with fatty acid composition, including 21 genes were annotated from fatty acid content SNPs (CCCH-type zinc finger family protein, glutamine-dependent asparagine synthase 1, arginyl-tRNA synthetase, class Ic, cullin 1 | NADH-ubiquinone oxidoreductase 24 kDa subunit, putative, plant U-box 24, uncharacterized protein family (UPF0497) | tetratricopeptide repeat (TPR)-like superfamily protein, response regulator 1, dicer-like 1, and NADPH-dependent thioredoxin reductase A) in mutant genotypes.

Of these, 11 SNPs from GBS were significantly associated with palmitic acid (C16:0) and located on chromosomes A2, A6, A8, C3, C5, C9, Ann, and Cnn. Four oleic acid (C18:1)-associated region were located on chromosomes A5, A7, and C9 in the mutant genotypes. We discovered 10 SNPs associated with linoleic acid (C18:2) located on chromosomes A3, A6, A9, and C5. Nine SNPs were associated with linolenic acid (C18:3) and located on chromosomes A5, A6, C3, C5, and C9. Three SNPs were associated with erucic acid (C22:1) and located on chromosomes A5, A10, and C9. Of the two other candidate SNP loci, both were associated with crude fat content, and one SNP was associated with flowering time. To explore whether the genes related to fatty acid composition were related to gene functions, we performed enrichment analysis. Enrichment analysis was performed by the 21 genes from 40 significant associated SNPs (Table 4) based on the threshold of FDR < 0.05. We found that nine genes were significantly enriched in reproductive processes

with sub-processes in the BP term (FDR < 0.05), such as embryonic development, fruit development, and seed developmental processes (Table 5).

**Table 4.** List of 40 significant associated SNPs with fatty acid and flowering types by association mapping.

| Triats | SNP | $-\log_{10}(p)$ | $R^2$ | Genic/Intergenic | TAIR ID | Allele |
|---|---|---|---|---|---|---|
| C16:0 | chrA02_4108741 | 4.45 | 0.187 | BnaA02g08450D | AT5G56930 | G/A |
| C16:0 | chrA06_12239044 | 4.18 | 0.174 | BnaA06g19730D | AT3G44330 | A/T |
| C16:0 | chrA07_4405396 | 4.29 | 0.242 | BnaA07g04220D | AT2G15530 | T/A |
| C16:0 | chrA08_18487790 | 4.02 | 0.191 | BnaA08g27990D | AT1G04650 | T/C |
| C16:0 | chrC03_39637954 | 4.43 | 0.239 | BnaC03g53730D | AT3G47340 | G/A |
| C16:0 | chrC05_23140515 | 4.39 | 0.35 | BnaC05g26850D | AT2G05642 | A/C |
| C16:0 | chrC09_7177749 | 4.68 | 0.222 | BnaC09g10630D | AT1G62200 | T/C |
| C16:0 | chrAnn_random_36154423 | 4.54 | 0.33 | Intergenic | | C/T |
| C16:0 | chrAnn_random_36154429 | 4.54 | 0.33 | Intergenic | | G/A |
| C16:0 | chrCnn_random_26697557 | 4.42 | 0.187 | Intergenic | | G/T |
| C16:0 | chrCnn_random_66012495 | 4.73 | 0.244 | Intergenic | | C/T |
| C18:1 | chrA05_11395334 | 4 | 0.296 | Intergenic | | C/T |
| C18:1 | chrA07_9257540 | 4.01 | 0.173 | BnaA07g09460D | AT1G26590 | T/C |
| C18:1 | chrC09_29915064 | 4.02 | 0.378 | Intergenic | | A/G |
| C18:1 | chrA07_random_856463 | 4.95 | 0.216 | BnaA07g37090D | AT1G26130 | G/A |
| C18:2 | chrA03_23022215 | 4.4 | 0.513 | BnaA03g45260D | AT4G22290 | T/C |
| C18:2 | chrA03_24661521 | 4.05 | 0.348 | BnaA03g47930D | AT4G26300 | G/A |
| C18:2 | chrA06_8948451 | 4.34 | 0.097 | BnaA06g16110D | AT3G49180 | G/A |
| C18:2 | chrA06_8948477 | 4.37 | 0.116 | BnaA06g16110D | AT3G49180 | G/A |
| C18:2 | chrA09_523273 | 4.73 | 0.432 | Intergenic | | A/G |
| C18:2 | chrA09_537232 | 4.39 | 0.263 | BnaA09g00890D | AT4G02570 | C/T |
| C18:2 | chrA09_537235 | 4.39 | 0.263 | BnaA09g00890D | AT4G02570 | C/T |
| C18:2 | chrA09_572373 | 4.18 | 0.008 | BnaA09g01000D | AT4G02700 | C/T |
| C18:2 | chrC05_39137705 | 4.44 | 0.16 | BnaC05g41370D | AT3G11840 | T/G |
| C18:2 | chrC05_39137834 | 4.44 | 0.16 | BnaC05g41370D | AT3G11840 | A/T |
| C18:3 | chrA05_19815348 | 4.32 | 0.107 | BnaA05g27500D | AT3G11550 | T/C |
| C18:3 | chrA05_19815372 | 4.32 | 0.107 | BnaA05g27500D | AT3G11550 | A/G |
| C18:3 | chrA06_3024875 | 4.53 | 0.185 | BnaA06g05280D | AT1G56330 | G/T |
| C18:3 | chrC03_47213635 | 4.24 | 0.004 | Intergenic | | T/C |
| C18:3 | chrC03_47213650 | 4.24 | 0.004 | Intergenic | | C/G |
| C18:3 | chrC03_47213704 | 4.54 | 0.003 | Intergenic | | A/G |
| C18:3 | chrC05_39137705 | 5.82 | 0.254 | BnaC05g41370D | AT3G11840 | T/G |
| C18:3 | chrC05_39137834 | 5.82 | 0.254 | BnaC05g41370D | AT3G11840 | A/T |
| C18:3 | chrC09_45061943 | 4.25 | 0.209 | BnaC09g44040D | AT5G13060 | C/T |
| C22:1 | chrA05_17476295 | 5.99 | 0.002 | BnaA05g23050D | AT3G16857 | T/C |
| C22:1 | chrA10_396151 | 8.15 | 0.002 | BnaA10g00800D | AT1G01040 | G/A |
| C22:1 | chrC09_6051633 | 4.11 | 0.002 | BnaC09g09340D | AT2G17420 | A/T |
| Crude | chrC08_17724544 | 4.11 | 0.178 | Intergenic | | T/C |
| Crude | chrCnn_random_26697557 | 4.62 | 0.099 | Intergenic | | G/T |
| FT | chrA09_19120040 | 4.08 | 0.291 | BnaA09g25970D | AT3G08570 | T/G/C |

**Table 5.** Summary of SNPs and genes to enrichment of reproductive process in BP term.

| Triats | SNP | Chr | Position | *B. napus* | TAIR Gene ID | TAIR Description |
|--------|-----|-----|----------|-----------|--------------|------------------|
| C16:0 | chrA02_4108741 | A02 | 4108741 | BnaA02g08450D | AT5G56930 | CCCH-type zinc finger family protein |
| C16:0 | chrC03_39637954 | C03 | 39637954 | BnaC03g53730D | AT3G47340 | glutamine-dependent asparagine synthase 1 |
| C18:2 | chrA03_24661521 | A03 | 24661521 | BnaA03g47930D | AT4G26300 | Arginyl-tRNA synthetase, class Ic |
| C18:2 | chrA09_537232 | A09 | 537232 | BnaA09g00890D | AT4G02570 | cullin 1 ∣ NADH-ubiquinone oxidoreductase 24 kDa subunit, putative |
| C18:2 | chrA09_537235 | A09 | 537235 | | | |
| C18:2 | chrC05_39137705 | C05 | 39137705 | | | |
| C18:2 | chrC05_39137834 | C05 | 39137834 | BnaC05g41370D | AT3G11840 | plant U-box 24 |
| C18:3 | chrC05_39137705 | C05 | 39137705 | | | |
| C18:3 | chrC05_39137834 | C05 | 39137834 | | | |
| C18:3 | chrA05_19815348 | A05 | 19815348 | BnaA05g27500D | AT3G11550 | Uncharacterised protein family (UPF0497) ∣ Tetratricopeptide repeat (TPR)-like superfamily protein |
| C18:3 | chrA05_19815372 | A05 | 19815372 | | | |
| C22:1 | chrA05_17476295 | A05 | 17476295 | BnaA05g23050D | AT3G16857 | response regulator 1 |
| C22:1 | chrA10_396151 | A10 | 396151 | BnaA10g00800D | AT1G01040 | dicer-like 1 |
| C22:1 | chrC09_6051633 | C09 | 6051633 | BnaC09g09340D | AT2G17420 | NADPH-dependent thioredoxin reductase A |

## 4. Discussion

Rapeseed cultivars have been improved in terms of seed-oil quality and yield. Industrial requirements for rapeseed oils are high-quality fatty acids, and this is recognized by various breeders [4–6]. Rapeseed oil is rich in mono- and poly-unsaturated fatty acids, such as oleic acid, linoleic acid, linolenic acid, and erucic acid; of these fatty acids, oleic and linoleic acid are generally recognized to be useful oil compounds [1,3]. However, linolenic acid is readily oxidized, which leads to decreased stability for high-temperature frying and shelf-life [1,5–7]. Therefore, a major goal has been breeding rapeseed with high oleic acid and/or low linolenic acid content. Rapeseed oil with higher oleic acid has many benefits compared with oil from low oleic acid cultivars, such as anti-oxidative properties, cost-effectiveness in fried cooking, and human health benefits [1–3]. Additionally, the consumption of oils with low levels of saturated fatty acids reduces the accumulation of low-density lipoprotein (LDL) cholesterol and the occurrence of heart disease in humans [3,4,7]. The average oleic acid contents of commercial rapeseed cultivars range from 60% to 65% of the total fatty acid content in Korea [41,42]. The oleic acid contents of the Tm6-8 and Tm6-13 mutant lines were higher ($\geq$75%) than those from the other mutant lines. In addition, Tm6-8 and Tm6-13 mutant lines had significantly lower levels of saturated fatty acids (palmitic and stearic acid) compared with the original cultivar. Pod shattering is another important technological trait in rapeseed as it is the main reason of grain loss. Tm2M-1 had the strongest to pod shattering resistance (data not shown). These results indicate that these mutant lines are potentially useful as materials for developing new rapeseed cultivars.

Recently, many rapeseed genetic resources are being used in horticulture, food and biodiesel industry [1–5]. In spite of its great industrial value, Korean has limited genetic resources for oil sources of rapeseed cultivars. The 'Tammi' cultivar has major agronomic characteristics such as early flowering, high seed yield (3420 kg/ha) and non-erucic acid oil [41]. The fatty acid composition of 'Tammi' seed oil is common types. Therefore, it is not highly valuable for the oil industry. Ionizing radiation mutagens have been widely applied to the study of genetics and mutation-based breeding programs [12,14,15]. Ionizing radiation contributes to free radical generation and causes DNA damage. It has been widely used in mutation breeding because it can generate useful traits in plant genetic resources [12,15,16,43]. Gamma rays are widely used as mutagens for plant-mutation breeding because of their accuracy and their deep penetration of plant organs that can cause mutations affecting chemical compositions [12,44]. Generally, an easy method for modifying oil traits using mutation is by inducing mutated genes involved in metabolic pathways that alter fatty acid composition in oil crops [5,14]. Modifications of the fatty acid composition in oil crops have been achieved beyond naturally-occurring variation by mutagenesis [12,15,17]. Gamma irradiation has been found to be an especially effective method of improving oil traits in plants [12]. Gamma induced mutation for altering fatty

acid composition has been the most frequently carried out and the mutants obtained had increased or decreased values of linolenic acid without any change in linoleic acid [12,45]. Kumar et al. [33] reported gamma rays induced Indian mustard mutant were developed containing a high oil percentage. Moreover, Rapeseed has been mutagenized using gamma rays for the improvement of several important characteristics, including reduction of toxin levels, such as glucosinolates and erucic acid [12,45]. In this study, oleic acid content has changed most and palmitic acid composition is lowest. Gamma irradiation can be used as an effective method for increasing the variability of oleic acid contents and lowering saturated fatty acids in oil crops [12,46]. Similar to the previous of development of high oleic acid composition (over 70%) of rapeseed breeding lines were obtained by 80-100 kR gamma ray treatment and oleic acid composition of $M_5$ progeny increased greatly [45]. However, Kim et al., reported that the wild-type soybean produced 9.4% palmitic acid, 2.3% stearic acid, 26.6% oleic acid, 53.0% linoleic acid, and 8.8% linolenic acid, while the gamma-induced mutant (Hfa180) produced 9.4% palmitic acid, 19.7% stearic acid, 19.4% oleic acid, 43.0% linoleic acid, and 8.5% linolenic acid [13]. The development of erucic acid-inducing mutants are unintentional product. Normally, the conventional mutagenesis approaches induce random mutations in the plant genome [12,14]. Similarly, the gamma irradiation induced mutation influence on oleic acid and erucic acid composition in rapeseed [45,46]. Those mutant lines can be used as a material to study for biosynthesis of erucic acid.

NGS techniques have been applied to the molecular research of rapeseed. GBS is the effective method for analyzing genetic variations conferred by large numbers of SNPs, and it is possible to use it to develop marker systems for crop selection due to its low cost and locus specificity [20–23]. The NGS techniques has led to the identification of mutations. When comparing the mutation breeding lines, such as induced mutant and its original cultivar, the only expected differences are the mutations caused by the mutagen [47]. However, there is a lack of research about using NGS techniques to study the mutations in fatty acids arising from gamma irradiation of rapeseed genetic resources. In this study, we discovered new SNPs in rapeseed mutant lines, including mutant lines related to flowering time and oil traits (crude fat content and fatty acid composition). In this study, we discovered new SNPs in rapeseed mutant lines, including mutant lines related to flowering time and oil traits (crude fat content and fatty acid composition). As result of GBS, a large number of SNPs were detected on chromosomes C01, C03, A03, and A09. The number of SNPs ranged from 2376 to 9446, with 6022 SNPs per mutant line on average. In GBS analysis, total amount of produced sequences may not be even among samples. Library preparation process contains DNA amplification step which may result in large differences in final outputs even when the amount of input DNA or quality of DNA is slightly different between samples [48]. Although there were substantial differences among samples in sequencing depth, the lowest average of sequencing depth among samples was much higher than $5\times$ which was the minimum threshold in SNP filtering. Therefore, there was not significant problem in SNP detection in samples with low sequencing depth [49,50]. In a previous rapeseed investigation of 633 ecotype germplasms, the GBS method detected 96.4 SNPs per sample, which was higher than the SNPs per sample detected in the present research [51]. Furthermore, there were more SNPs in the 38 mutant lines compared with the reference genome sequence. These results suggested that gamma ray treatment could effectively induce genetic variability in rapeseed plants.

Ultimately, we detected 35,397 high-quality SNPs, which were used for phylogenetic analysis. In this study, we found high levels of genetic diversity among the rapeseed mutant lines derived from gamma irradiation. The phylogenetic tree showed that all mutant lines were classified regardless of their flowering time and crude fat and fatty acid contents. Accordingly, the classification of genetic diversity and variation is one of the most important elements in the selection of germplasms for breeding programs and gives other useful information [52,53]. Hierarchical cluster analysis categorized the 47 rapeseed mutant lines according to their flowering times and/or oleic acid levels. These trends suggest that the oleic acid content of rapeseed oil could be used as a marker to assess

chemotypes. Consequently, hierarchical cluster analysis is important for evaluating the chemical diversity of rapeseed as well as genetic variation among novel mutant lines. There was no agreement between the chemical and genetic classifications. Overall results could be applied to breeding programs to develop rapeseed cultivars with improved flowering times and fatty acid compound profiles.

We conducted GO analysis of rapeseed genes containing polymorphic SNPs induced by gamma radiation. The GO grouping of mutated genes indicated that many variations induced functional changes. For the categories of BP, mutations occurred most frequently in genes involved in cellular processes, primary metabolic processes, and nitrogen compound metabolic processes. Of the CC categories, intracellular entities were the major term, followed by organelles, and intracellular organelles. For the MF categories, the major polymorphic SNPs were nucleotide binding, nucleoside phosphate binding, anion binding, and carbohydrate derivative binding. Gamma rays can affect plant cell metabolism through thylakoid membrane dilation, alteration of photosynthesis, and changes to secondary metabolism [12,14–16]. Previously, Go analysis of variation in *Dendrobium* mutant derived from gamma ray, aerospace, and somaclonal mutagenesis, including cellular aromatic compound metabolic (BP), nucleobase-containing compound metabolic process (BP), protein metabolic process (BP), intracellular membrane-bound organelles (CC), nucleic acid binding (MF), cation binding(MF) and nucleotide binding (MF) [54]. While, GO analysis revealed that genes of rapeseed, including response to herbivore, metabolic process, regulation of immune response, axillary shoot meristem initiation, meristem initiation, triglyceride biosynthetic process, terpenoid metabolic process, development process, photomorphogenesis, trichome morphogenesis, regulation of cell proliferation, and inflorescence development exhibited high mutation rate in 588 natural populations [55]. Therefore, it appears believable to expect that an appreciable difference in mutations is observed between natural variation and radiation mutagenesis.

Of the 35,397 high-quality SNPs, 40 SNPs were significantly associated with flowering time, crude fat content, and fatty acid composition. We found SNPs associated with flowering time, including one in an NPH3 family/ATP synthase subunit on chromosome A09. Rapeseed flowering time affects its yield, cultivation area, and ornamental value. Previous reports indicate that rapeseed flowering time is controlled by complex factors through allelic effects [25]. In rapeseed, the co-localization of a flowering time QTL with yield and SNP marker density was sufficient to describe several candidate genes on chromosomes A03, C04 and C08 [25,32]. The high oil yield of rapeseed may be due to improvements in the application of large amounts of fertilizers and pesticides, which may negatively impact the farmer. The development of new rapeseed cultivars with high crude fat contents could solve this problem [2,12,46]. We observed that two SNPs significantly associated with crude fat content were found on chromosomes C08 and Cnn. Unfortunately, these SNPs occur in the intergenic region. Lu et al. reported that chromosome A-specific selection may have promoted the oil accumulation among the landrace of rapeseed population [55]. It also believable to expect that an appreciable difference in mutation is radiation mutagenesis. We observed that 37 SNPs were significantly associated with fatty acid composition. Specifically, these SNPs occur in nine genes involved in embryonic development, fruit development, and seed development in our study (Figure S1). Seed fatty acid biosynthetic pathways have been identified in plants. Fatty acid biosynthetic pathways are related to specific enzymes, which form a cooperative system for seed development, energy metabolism, and triacylglycerol biosynthetic pathways [28,29]. Additionally, the marker associated technique has been applied in the identification of genetic variation of inbred lines, including flowering time, pathogen resistance, glucosinolate content, and phenolic compounds in rapeseed [29–32,56,57]. The SNPs related to flowering time, crude fat content and fatty acid compositions could be developed as a marker for selecting lines.

The fatty acid desaturase 2 (FAD2) and fatty acid desaturase 3 (FAD3) genes encode key enzymes responsible for producing precursors of oleic and/or linoleic acid in the canola lipid biosynthetic pathway [5,28,29]. Rapeseed FAD2 genes are separated and

located on chromosomes A01, A05, C01, and C05, and they may be involved in oleic acid regulation [29,31,32]. In our study, a total of nine SNPs were detected on chromosomes A05 and C05, including plant U-box 24, uncharacterized protein family (UPF0497) | tetratricopeptide repeat (TPR)-like superfamily protein, and response regulator 1. The linolenic acid content in rapeseed is controlled by FAD3 genes; they encode delta-15 linoleate desaturase, which is responsible for the dehydration of linoleic acid to form linolenic acid. Furthermore, we observed 19 SNPs significantly associated with linoleic and linolenic acid content on chromosomes A03, A05, A06, A09, C03, C05, and C09. However, most of the SNPs significantly associated with linoleic and linolenic acid content were found on chromosomes A08 and C03, and their detection varied widely depending on the environment [29–32]. Additionally, 30 SNP loci significantly associated with erucic acid, oleic acid, and linoleic acid contents were detected by specific locus amplified fragment sequencing and identified functional candidate genes related to fatty acid biosynthesis, including FAE1, FAD2, LACS09, KCS17, CER4, TT16, and ACBP5 on chromosomes A06, A08, A09, A10, C02, C03, C07, and C08 in different environments [26]. Fatty acid content is a typical quantitative trait controlled by multiple genes that regulate fatty acid desaturation, and most genes residing on chromosomes A03, A05, A06, A10, and C02 were repeatedly detected due to natural variation in rapeseed [26,31,55]. Moreover, an ethyl methanesulfonate (EMS)-treated population with varying fatty acid contents was also investigated; high oleic acid content mutants had nucleotide deletions in FAD2 on chromosomes A01 and A05 [19]. Previously, two genomic regions on chromosome C3 were associated with QTL for six (palmitic, stearic, oleic, linoleic, linolenic and archidic acid) fatty acid content [58] and a total of 406 identified QTLs were detected for eight fatty acid content in China and 14 microenvironments were involved [59]. Furthermore, 101 consensus QTLs were detected for six fatty acid using an inbreeding lines in six environments [60]. Identification of agronomical important genes are crucial benefits for plant breeding [61]. Identifying the genetic variation between SNP loci and mutants will be beneficial for improving the oil qualities of rapeseed. Therefore, the candidate genes selected in this study should be verified by additional analysis.

## 5. Conclusions

In conclusion, GBS was used to develop high-quality SNPs in rapeseed mutant genotypes that were subsequently used to assess genetic diversity and variation. Novel SNP locations for flowering time, crude fat content, and fatty acid composition in the rapeseed mutant lines were found by an association study. We revealed that GBS technology is a powerful tool for identifying SNPs for rapeseed mutants derived by gamma irradiation. Based on GBS and FarmCPU model analysis, significant association signals for flowering time, crude fat content, and palmitic, oleic, linoleic, linolenic, and erucic acids in seeds of rapeseed were found on chromosomes A02, A03, A05, A06, A07, A09, A10, Ann, C03, C05, C08, C09, and Cnn. Specifically, these results indicate that significantly associated SNPs are reliably involved in fatty acid biosynthesis and metabolism during seed reproductive stages in our study. This is the first report describing SNPs generated by GBS for mutants derived by gamma irradiation. Our results may improve molecular breeding for suitable SNPs among rapeseed mutants and could be useful for the development of novel cultivars for mutation breeding programs.

**Supplementary Materials:** The following are available online at https://www.mdpi.com/2073-4395/11/3/508/s1, Table S1. Results of genotyping-by-sequencing analysis of rapeseed mutant lines, Table S2. Summary of GBS sequence data and alignment to the reference genome sequence, Table S3. List of union SNP matrix loci that were generated for 46 rapeseed mutant lines. Table S4. List of functional groups under the three main categories; cellular components, biological process, and molecular function. Figure S1. Diagram of the significant enriched term of GOBP by candidate SNPs to enrichment of reproductive process in BP term. Supplementary Materials File S1: VCF file of union SNP matrix.

**Author Contributions:** Conceptualization: J.-W.A. and S.-Y.K.; formal analysis: J.R., J.I.L., D.-G.K., K.M.K., B.Y., Y.D.J. and S.H.K.; investigation: J.R., J.I.L., J.-B.K., S.-J.K. and B.-K.H.; writing—original draft preparation: J.R., J.I.L., K.M.K. and S.H.K.; writing—review and editing: J.-B.K., J.-W.A. and B.-K.H.; project administration: J.-B.K.; funding acquisition: J.-W.A. All authors have read and agreed to the published version of the manuscript.

**Funding:** This work was supported by the research program of KAERI, Republic of Korea and the Radiation Technology R&D Program (NRF-2017M2A2A6A05018538) through the National Research Foundation of Korea funded by the Ministry of Science and ICT.

**Institutional Review Board Statement:** Not applicable.

**Informed Consent Statement:** Not applicable.

**Data Availability Statement:** Not applicable.

**Conflicts of Interest:** The authors declare that they have no conflict of interest.

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
