# Peer review of "Single Nucleotide Polymorphism (SNP) Discovery and Association Study of Flowering Times, Crude Fat and Fatty Acid Composition in Rapeseed (Brassica napus L.) Mutant Lines Using Genotyping-by-Sequencing (GBS)"

_agronomy, doi:10.3390/agronomy11030508_

Round 1

Reviewer 1 Report

The authors present an association study using SNPs from GBS of rapedded mutant lines, which were mutated by means of gamma rays, seeking an increase in their crude fat and fatty acids.

Minor observations are suggested.

In the introduction expand more on the characteristics of the plant, for example:

Rapessed is an oilseed plant that grows three to five feet tall and produces beautiful little yellow flowers, for example.

The rapessed plant is part of the brassica family, as are cabbage, broccoli, Brussels sprouts, and mustard.

The plant produces pods from which seeds are harvested. Rapessed oil is obtained from ground seeds. These seeds contain about XX percent oil, more than half that of …… (another oil-producing grain) ……. Etc.

Line 41
Indicate why it is considered harmful to health

Lines 49.50
"However, a demand for rapeseed oils with other fatty acid
 compositions exists in the market ".
Which ones and why is there this demand?

Line 292
It should be figure 2.

Figure 2
Indicate the name of the reference assembly and its version.

Line 400, This is already mentioned in line 41, delete.
Furthermore, erucic acid appears to have toxic effects on humans and animals.

Lines 426-429
They mention similar studies, but do not compare any results of their study against the other reported studies, I suggest they discuss results between these studies and mention the most significant results / differences.

Lines 445-446
Discuss what they infer that these chromosomes contain the largest number of SNPs and those chromosomes to which they are associated.

Lines 546-547
How much different?

Discuss and add in the discussion chapter:

derived from the mutation how much fatty acids increased?

Which was the fatty acid that increased the most or which one decreased the most?

What percentage of SNPs were associated with oil-producing genes?

Show conclusion as chapter 5.

Author Response

Dear reviewer 1 
On behalf of my co-authors, we thank you for giving us an opportunity to revise our manuscript, and we appreciate the editor and reviewers for your positive and constructive comments and suggestions on our manuscript entitled; Single Nucleotide Polymorphism (SNP) Discovery and Association Study of Flowering times, Crude fat and Fatty acid Composition in Rapeseed (Brassica napus L.) Mutant Lines Using Genotyping-by-Sequencing (GBS)
We have reviewed our manuscript carefully, and tried our best to improve our manuscript and made some changes in the manuscript based on the editor and reviewers’ comments. We marked the changes in red in our revised paper. We appreciate for editor and reviewers’ warm work, and hope that the correction will meet with approval. The main corrections in our manuscript and the responds to the reviewer’s comments are as follows:
REVIEWER COMMENTS:   
The authors present an association study using SNPs from GBS of rapedded mutant lines, which were mutated by means of gamma rays, seeking an increase in their crude fat and fatty acids. Minor observations are suggested.
[Response] Thank you

1.    In the introduction expand more on the characteristics of the plant, for example:

Rapessed is an oilseed plant that grows three to five feet tall and produces beautiful little yellow flowers, for example.
The rapessed plant is part of the brassica family, as are cabbage, broccoli, Brussels sprouts, and mustard.
The plant produces pods from which seeds are harvested. Rapessed oil is obtained from ground seeds. These seeds contain about 25 to 50  percent oil, more than half that of …… (another oil-producing grain) ……. Etc.
[Response] We thank the reviewer for this comment. We have added (Line 34-39)

2.    Line 41 Indicate why it is considered harmful to health
[Response] We thank the reviewer for this comment. We have added (Line 46-47)

3.    Lines 49.50 "However, a demand for rapeseed oils with other fatty acid
 compositions exists in the market ".Which ones and why is there this demand?
[Response] As suggested, we add to information (line 56-57) 

4.    Line 292 It should be figure 2.
[Response] We apologize sorry. We changed it
5.    Figure 2 Indicate the name of the reference assembly and its version.
[Response] As suggested, we add to assembly information in Figure 2 

6.    Line 400, This is already mentioned in line 41, delete. Furthermore, erucic acid appears to have toxic effects on humans and animals.
[Response] We thank the reviewer for this comment. We have deleted and added information (Line 412 and 46-47)

7.    Lines 426-429
They mention similar studies, but do not compare any results of their study against the other reported studies, I suggest they discuss results between these studies and mention the most significant results / differences.
[Response] We thank the reviewer for this comment. We have added information (Line 453-459 and line 461-463)

8.    Lines 445-446
Discuss what they infer that these chromosomes contain the largest number of SNPs and those chromosomes to which they are associated.
[Response] We apologize sorry to confuse. These were indicated the distribution of total SNPs from GBS, which are not related to fatty acids and flowering. To avoid confusion, we have changed these sentence as follows: “In this study, we discovered new SNPs in rapeseed mutant lines, including mutant lines related to flowering time and oil traits (crude fat content and fatty acid composition). As result of GBS, a lot of SNPs were detected on chromosomes C01, C03, A03, and A09. The number of SNPs ranged from 2376 to 9446, with 6022 SNPs per mutant line on average.” (Line 474-478)

9.    Lines 546-547 How much different?
[Response] We apologize sorry. We cannot confirm all relevant previous studies. We have deleted sentence and added references (Line 574-578)
Discuss and add in the discussion chapter:

10.    derived from the mutation how much fatty acids increased?
[Response] It varies greatly depending on the mutant lines. This is related to number 11.
11.    Which was the fatty acid that increased the most or which one decreased the most? 
[Response] We thank the reviewer for this comment. We have added information (Line 450-451)
12.  

What percentage of SNPs were associated with oil-producing genes?
[Response] We apologize sorry. We cannot confirm all oil-producing genes. It because the oil synthesis is very complex.

Show conclusion as chapter 5.
[Response] We thank the reviewer for this comment. We have changed it.

Reviewer 2 Report

Title

In my opinion the title should be more concise and more precisely describe the main focus of this study, which is association of SNPs with fatty acid composition and flowering time, e.g. “Genotyping-by-Sequencing (GBS) of single nucleotide polymorphism (SNP) associated with fatty caid composition and flowering time in rapeseed (Brassica napus L.) mutant lines”. The above example is only my suggestion, however the title should be more descriptive.

Materials and methods

Subheading 2.3 - please provide accession/reference for the reference genome of B. napus.

Lines 152-153: If I understand correctly, the reference genome of B. napus was used for mapping of sequenced reads. Could you explain the term “standard dielectric” in this sentence?

Lines 173-174: Please clarify whether the “read depth” or “read rate” were used to classify the SNPs as homozygous or heterozygous?

Lines 210-212: This sentence is not clear, please correct it and rephrase.

Line 228: “a subsequent increase to 28° C” Please check if this temp. is correct or should it be “decrease” instead of “increase”?

Results

The terms “composition” and “content” should be used appropriately in the entire manuscript. The fatty acid composition refers to different fatty acids. If one particular fatty acid is mentioned use “content” instead of composition, e.g. “The palmitic acid content of the mutant lines ranged form…”  

Lines 254-256 and Table 1: please check if the if the content of erucic acid is correct. According to Table 1 its range is from 0.0 to 20.1. The highest content is in line Tm_10-1St, while in line Tm4M-2 is 0.0.

Table 1: please provide units  of crude fat [mg/100 g] and fatty acids [%] in the table heading.

Lines 270-271: The numbers of reads provided here and in Table 2 are inconsistent, please correct.

Line 286: It should be more clearly explained what authors mean by the “common SNP”. Is it a SNP shared between mutant and wild-type cultivar but not with the reference genome?

Table 5: please make the title of Table 5 more explanatory. Also if SNPs were confirmed, they cannot be “candidate”.

Discussion

Please shortly explain why this particular cultivar (Tammi) was chosen in this study.

Authors should discuss the novelty and significance of their results by referring to the previous studies of oil composition in rapeseed. There are several studies of QTLs associated with fatty acid composition in rapeseed, e.g. Javed et al. 2016 (DOI 10.1007/s10681-015-1565-2), Chen et al. 2018 ( https://doi.org/10.1186/s12870-018-1268-7), Bao et al. 2018 (https://doi.org/10.3389/fpls.2018.01018). It is worth to know if the SNPS discovered in this study overlap with the previously identified QTLs or are they unique and novel loci?

Pod shattering is another important technological trait in rapeseed as it is the main reason of grain loss. Although there were differences in flowering time between mutant lines, there is no information if these changes affected the resistance to pod shattering or final grain yield. At least short information if these traits were determined in mutant lines should be included in the discussion.

Lines 403-404 and 454-455: these sentences are not clear, please rephrase them.

General comments

The manuscript requires some grammar and style corrections.

Author Response

Dear reviewer2 
On behalf of my co-authors, we thank you for giving us an opportunity to revise our manuscript, and we appreciate the editor and reviewers for your positive and constructive comments and suggestions on our manuscript entitled; Single Nucleotide Polymorphism (SNP) Discovery and Association Study of Flowering times, Crude fat and Fatty acid Composition in Rapeseed (Brassica napus L.) Mutant Lines Using Genotyping-by-Sequencing (GBS)
We have reviewed our manuscript carefully, and tried our best to improve our manuscript and made some changes in the manuscript based on the editor and reviewers’ comments. We marked the changes in red in our revised paper. We appreciate for editor and reviewers’ warm work, and hope that the correction will meet with approval. The main corrections in our manuscript and the responds to the reviewer’s comments are as follows:
REVIEWER COMMENTS:   

Reviewer#2
1.    In my opinion the title should be more concise and more precisely describe the main focus of this study, which is association of SNPs with fatty acid composition and flowering time, e.g. “Genotyping-by-Sequencing (GBS) of single nucleotide polymorphism (SNP) associated with fatty caid composition and flowering time in rapeseed (Brassica napus L.) mutant lines”. The above example is only my suggestion, however the title should be more descriptive.

[Response] We thank the reviewer for this comment. We have added it 
Single Nucleotide Polymorphism (SNP) Discovery and Association Study of Flowering times, Crude fat and Fatty acid Composition in Rapeseed (Brassica napus L.) Mutant Lines Using Genotyping-by-Sequencing (GBS)

Materials and methods

2.    Subheading 2.3 - please provide accession/reference for the reference genome of B. napus.

 [Response] We thank the reviewer for this comment. We have added it (Line 159)

3.    Lines 152-153: If I understand correctly, the reference genome of B. napus was used for mapping of sequenced reads. Could you explain the term “standard dielectric” in this sentence?
[Response] In response to the reviewer’s opinion, we have changed “standard dielectric” to “B. napus V5.1” (Line 161)

4.    Lines 173-174: Please clarify whether the “read depth” or “read rate” were used to classify the SNPs as homozygous or heterozygous?
[Response] We apologize sorry to confuse. This is the correct “read depth”. We have changed word in Line 181.

5.    Lines 210-212: This sentence is not clear, please correct it and rephrase.
[Response] We thank the reviewer for this comment. We have changed it (line 218-221)
The annotated genes as selected significant SNPs were evaluated enrichment analysis of GOBP category for defined gene functions by AgriGO database (http://bioinfo.cau.edu.cn/agriGO/) based on the threshold of Hochberg multitest adjustment method [false discovery rate (FDR) < 0.05].

6.    Line 228: “a subsequent increase to 28° C” Please check if this temp. is correct or should it be “decrease” instead of “increase”?

[Response] We apologize sorry to confuse. We have changed word in Line 237.

Results

7.    The terms “composition” and “content” should be used appropriately in the entire manuscript. The fatty acid composition refers to different fatty acids. If one particular fatty acid is mentioned use “content” instead of composition, e.g. “The palmitic acid content of the mutant lines ranged form…”  

Response] We thank the reviewer for this comment. We have changed it (line 254-257) 

8.    Lines 254-256 and Table 1: please check if the if the content of erucic acid is correct. According to Table 1 its range is from 0.0 to 20.1. The highest content is in line Tm_10-1St, while in line Tm4M-2 is 0.0.
[Response] We apologize sorry to confuse. We have changed it (Line 263-265).

9.    Table 1: please provide units  of crude fat [mg/100 g] and fatty acids [%] in the table heading.

 Response] We thank the reviewer for this comment. We have changed it (Table 1) 

10.    Lines 270-271: The numbers of reads provided here and in Table 2 are inconsistent, please correct.
[Response] We thank the reviewer for this comment. We have changed “60,784,210,724” to “646,640,540” (Line 280)

11.    Line 286: It should be more clearly explained what authors mean by the “common SNP”. Is it a SNP shared between mutant and wild-type cultivar but not with the reference genome?
[Response] Yes, “common SNP” is different from the reference genome, and some may be the same as the wild-type cultivar. So, we has 2nd collecting process for identifying polymorphic SNPs compared with wild-type cultivar. To clarify, we some added information in this sentence as follows: “common SNPs (1st selected 1,528,875 SNPs)” (Line 295)  

12.    Table 5: please make the title of Table 5 more explanatory. Also if SNPs were confirmed, they cannot be “candidate”.
[Response] As suggested, we removed the “candidate” and more detailed explanatory in Table 5 as follows: Table 5. Summary of SNPs and genes to enrichment of reproductive process in BP term. Enrichment analysis was performed by the 21 genes from 40 significant associated SNPs (Table 4) based on the threshold of FDR < 0.05. (Line 339-401)

Discussion

13.    Please shortly explain why this particular cultivar (Tammi) was chosen in this study.

 [Response] We thank the reviewer for this comment. We have added it (Line 430-433)

14.    Authors should discuss the novelty and significance of their results by referring to the previous studies of oil composition in rapeseed. There are several studies of QTLs associated with fatty acid composition in rapeseed, e.g. Javed et al. 2016 (DOI 10.1007/s10681-015-1565-2), Chen et al. 2018 ( https://doi.org/10.1186/s12870-018-1268-7), Bao et al. 2018 (https://doi.org/10.3389/fpls.2018.01018). It is worth to know if the SNPS discovered in this study overlap with the previously identified QTLs or are they unique and novel loci?

[Response] We thank the reviewer for this comment. We have discussed it (Line 574-578)

Pod shattering is another important technological trait in rapeseed as it is the main reason of grain loss. Although there were differences in flowering time between mutant lines, there is no information if these changes affected the resistance to pod shattering or final grain yield. At least short information if these traits were determined in mutant lines should be included in the discussion.

 [Response] We thank the reviewer for this comment. We have added it (Line 423-425)

Lines 403-404 and 454-455: these sentences are not clear, please rephrase them.

[Response] We thank the reviewer for this comment. We have changed it (Lines 403-404; Line 413-414 and 454-455; deleted)

General comments

The manuscript requires some grammar and style corrections.
[Response] We thank the reviewer for this comment. We have English editing
